# Impact of smoke-free ordinance strength on smoking prevalence and lung cancer incidence

Ryan H. Nguyen[1]¤*, Laura B. Vater[1], Lava R. Timsina[2], Gregory A. Durm[1,3], Katelin Rupp[4], Keylee Wright[4], Miranda H. Spitznagle[4], Brandy Paul[4], Shadia I. Jalal[1,3], Lisa Carter-Harris[5], Karen S. Hudmon[3,6], Nasser H. Hanna[1,3], Patrick J. Loehrer[1,3], DuyKhanh P. Ceppa[2,3]

**1** Department of Internal Medicine, Indiana University School of Medicine, Indianapolis, IN, United States of America, **2** Department of Surgery, Indiana University School of Medicine, Indianapolis, IN, United States of America, **3** Indiana University Health Simon Cancer Center, Indianapolis, IN, United States of America, **4** Indiana State Department of Health, Indianapolis, IN, United States of America, **5** Memorial Sloan Kettering Cancer Center, New York, NY, United States of America, **6** Department of Pharmacy Practice, Purdue University College of Pharmacy, West Lafayette, IN, United States of America

¤ Current address: Department of Medicine, University of Illinois at Chicago, Chicago, IL, United States of America

* rnguye8@uic.edu

**Data Availability Statement:** All relevant data are within the paper and its Supporting Information files.

## Abstract

### Background

Smoke-free ordinances (SFO) have been shown to be effective public health interventions, but there is limited data on the impact SFO on lung cancer outcomes. We explored the effect of county-level SFO strength with smoking prevalence and lung cancer incidence in Indiana.

### Methods

We obtained county-level lung cancer incidence from the Indiana State Cancer Registry and county-level characteristics from the Indiana Tobacco Prevention and Cessation Commission's policy database between 1995 and 2016. Using generalized estimating equations, we performed multivariable analyses of smoking prevalence and age-adjusted lung cancer rates with respect to the strength of smoke-free ordinances at the county level over time.

### Results

Of Indiana's 92 counties, 24 had a SFO by 2011. In 2012, Indiana enacted a state-wide SFO enforcing at least moderate level SFO protection. Mean age-adjusted lung cancer incidence per year was 76.8 per 100,000 population and mean smoking prevalence per year was 25% during the study period. Counties with comprehensive or moderate SFO had a smoking prevalence 1.2% (95% CI [-1.88, -0.52]) lower compared with counties with weak or no SFO. Counties that had comprehensive or moderate SFO also had an 8.4 (95% CI [-11.5, -5.3]) decrease in new lung cancer diagnosis per 100,000 population per year compared with counties that had weak or no SFO.

**Funding:** The authors received no specific funding for this work.

**Competing interests:** No authors have competing interests.

## Conclusion

Counties with stronger smoke-free air ordinances were associated with decreased smoking prevalence and fewer new lung cancer cases per year. Strengthening SFO is paramount to decreasing lung cancer incidence.

## Introduction

Since the Surgeon General's first report on the health consequences of smoking in 1964, advances in public health policy have led to significant decreases in the prevalence of smoking and the incidence of tobacco-related diseases. However, tobacco use continues to be the leading known cause of preventable death in the United States, with lung cancer being the leading cause of cancer-related mortality in the nation [1]. While smoking causes the majority of lung cancer cases, second-hand smoke (SHS) has also been shown to cause lung cancer in adults who have never smoked [2].

The burden of tobacco-related disease weighs heavily on the State of Indiana, where at least 11,000 deaths and nearly $3 billion dollars of healthcare costs per year are attributable to smoking [3, 4]. Indiana exhibits one of the highest adult smoking prevalences and lung cancer incidence rates in the nation [5]. In 2015, an estimated 20.6% of adults were current smokers, and the lung cancer incidence was 72.3 per 100,000 population [5, 6]. On the other hand, Indiana remains below the national average in terms of cigarette taxation and tobacco prevention efforts. Since 2000, Indiana has raised the state cigarette tax from 15.5 to its current value (since 2007) of 99.5 cents per pack, compared with the national state average of $1.81 per pack in June 2019. In addition, in the fiscal year 2018, Indiana allocated $7.5 million on tobacco prevention, only 10.2% of the recommended budget for state tobacco-control programs by the Centers for Disease Control and Prevention [7].

In an effort to decrease tobacco-related morbidity and mortality, Indiana counties began implementing municipal smoke-free policies, starting with Monroe County in 2005. However, until 2011, only 24 of Indiana's 92 counties had any smoke-free ordinances (SFOs) [7]. These SFOs varied in strength of coverage from comprehensive (100% smoke-free workplaces and enclosed public places) to moderate (protection in workplaces and restaurants) to weak (protection in workplaces or some public places with significant exceptions such as separate smoking rooms). In 2012, Indiana enacted a statewide smoke-free law that mandated at least moderate SFO protections, with the option to voluntarily enforce stricter protections.

SFOs are one of the most effective public policy means of decreasing smoking prevalence and SHS exposure [8–10]. Comprehensive SFOs that cover all workplaces and public places are associated with decreased risks of pulmonary and cardiovascular disease [11–13]. However, limited data are available on the relationship between smoke-free policies and lung cancer outcomes. In this study, we compared lung cancer incidence rates and smoking prevalence among Indiana counties based on strength of SFO. By including the period following the passage of the statewide smoke-free law, we also sought to examine the effect of expanding smoke-free protections to counties with weaker or no previous SFOs. We hypothesized that stronger SFOs would be associated with decreased smoking prevalence and lung cancer incidence rates.

## Materials and methods

### Smoke-free ordinances

Accessing the smoke-free ordinance database at the Indiana Tobacco Prevention and Cessation Commission, we obtained information about each smoke-free policy implemented in

Indiana from 1995–2016, including the date and strength of the policy (i.e., comprehensive, moderate, or weak). Municipal SFOs granted smoke-free protections to either entire counties or cities that were typically the largest city in the county. Consistent with methodology from prior studies, we classified counties as having the same level of SFO coverage as a city within the county that passed the SFO.

## Lung cancer incidence

Querying the Indiana State Cancer Registry (ISCR) database from 1995 through 2016, we identified 110,935 newly diagnosed cases of lung cancer among Indiana residents. The ISCR, which is part of the National Cancer Institute's Surveillance, Epidemiology, and End Results program and the Center for Disease Control and Prevention's National Program of Cancer Registries, is the official population-based cancer reporting system and includes all reported cases of primary malignant diseases diagnosed or treated at an Indiana hospital. For each unique patient case, we received demographic data and case-specific information (e.g., site code, date of diagnosis, tumor stage). County-level annual age-adjusted lung cancer incidence rates were calculated utilizing this dataset.

## County-level characteristics

County-level demographics included the adult smoking prevalence, average population, median household income, percentage below the poverty line, percentage high school graduates, percentage with bachelor's degree, percentage white, percentage black, percentage female, and location type (rural/urban). The annual adult smoking prevalence, calculated as the 5-year average from the Behavioral Risk Factor Surveillance System (BRFSS), was included as a time-dependent covariate (recorded yearly). Because the annual number of BRFSS participants in smaller counties did not meet the threshold for smoking rate estimation and consistent county-level data were not available until 2000, weighted 5-year averages from 2000 through 2015 were used. County estimates for the average population, median household income, education levels, percentage below the poverty line, percentage high school graduates, percentage with bachelor's degrees, race/ethnicity, and sex were provided by the 2000 and 2010 U.S. census. The 2000 Rural Urban Continuum codes were used to assign location type (urban or rural), with codes from 1 to 3 indicating urban locations and codes from 4 to 9 indicating rural locations.

## Statistical analysis

Using generalized estimating equations (GEE) with robust standard errors and accounting for the clustering effect at county level, we performed multivariable analyses of smoking prevalence and age-adjusted lung cancer rates with-respect-to the strengths of smoke-free ordinances at the county level over time. The county level characteristics over time served as control variables in each GEE model. This model allowed flexibility to modify the indicator for strength of municipal SFOs to reflect policy changes that occurred over time. These included changing from no ordinances to a municipal ordinance or strengthening of initial ordinance after implementation. Estimates of differences between smoking prevalence and lung cancer incidence between counties and corresponding 95% confidence intervals were calculated for each model effect, and significance levels for the estimates were determined using Wald tests. We conducted our analysis with Stata/SE 14.2 [StataCorp. 2015. Stata Statistical Software: Release 14. College Station, TX: StataCorp LP] software.

Our study was granted exemption status by the Indiana University Institutional Review Board. All data samples were fully anonymized before we accessed them.

## Results

From 1995 to 2016, a total of 110,935 newly diagnosed cases of lung cancer in Indiana residents were recorded (Table 1). Males comprised 56% of cases, and whites (92.5%) and African-Americans (7.1%) made up the majority of cases. The largest represented age groups were 65–74 years (34.7%), >75 years (31.5%), and 50–64 (28.5%). Nearly half (49.6%) were Stage III or IV lung cancers and 33% were not staged.

The median county-level population was 33,812 (range: 5,876 to 881,924), and average median household income among counties was $45,548 (range: $35,976 to $82,054) (Table 2). The average percentage of African-Americans in counties was 2.5%, with a wide range from 0.1% to 26.7%. The average percentage of those living under the poverty line was 13.9% (range: 4.9% to 24.3%), and 16.8% held an undergraduate degree (range: 7.5% to 53.8%). Fifty percent of counties were located in metropolitan locations.

Fig 1 demonstrates changes in SFO strength over time. The first municipal SFO was passed in 2005, and up until 2011, only 24 of Indiana's 92 counties had passed SFOs of any strength. Of these ordinances, twelve (50%) were comprehensive, eight were moderate (33.3%), and four were weak (16.7%). The 2012 Indiana Smoke-Free Air Law covered the entire state and effectively converted the 68 counties without any SFOs and 4 counties with weak SFOs into moderate SFO counties. From 2013–2016, one county elected to increase the strength of its moderate SFO to comprehensive. By 2016, 77 counties (83.7%) were protected by moderate SFOs, and 15 counties (16.3%) had comprehensive SFOs.

**Table 1. Patient demographics of Indiana residents diagnosed with lung cancer (n = 110,935).**

| Characteristics | | Overall (%) | 1995–2012 (%) | 2012–2016 (%) | p-value |
|---|---|---|---|---|---|
| Age group (years) | | | | | <0.01 |
| | ≤49 | 5.2 | 5.7 | 4 | |
| | 50–64 | 28.5 | 28.2 | 29.9 | |
| | 65–74 | 34.7 | 34.9 | 34.5 | |
| | 75+ | 31.5 | 31.3 | 31.6 | |
| Gender | | | | | <0.01 |
| | Male | 56.0 | 56.8 | 53.4 | |
| | Female | 44.0 | 43.2 | 46.6 | |
| Race | | | | | <0.01 |
| | White | 92.5 | 92.5 | 92.5 | |
| | Black or African American | 7.1 | 7.1 | 7 | |
| | American Indian and Alaska Native | 0.0 | 0.0 | 0.1 | |
| | Asian or Pacific Islander | 0.3 | 0.3 | 0.3 | |
| | Other | 0.1 | 0.1 | 0.1 | |
| | Unknown | 0.1 | 0.1 | 0.1 | |
| Ethnicity | | | | | <0.01 |
| | Hispanic | 97.1 | 96.4 | 99 | |
| | Non-Hispanic | 0.7 | 0.6 | 0.7 | |
| | Unknown | 2.3 | 2.9 | 0.3 | |
| Clinical stage | | | | | <0.01 |
| | 0 | 0.2 | 0.1 | 0.2 | |
| | I | 13.2 | 11.1 | 20.6 | |
| | II | 4.2 | 3.5 | 6.9 | |
| | III | 17.6 | 17.4 | 18.8 | |
| | IV | 32 | 28.8 | 43.4 | |
| | Unstaged | 33 | 39.1 | 10.1 | |

**Table 2. Indiana county-level characteristics (counties: n = 92).**

| Characteristics | Mean | Std Dev | Min | Max |
|---|---|---|---|---|
| Population | 68,284.5 | 113,160.3 | 5876 | 881,924 |
| Median income (USD) | 45,548.2 | 7,372.4 | 35,976 | 82,054 |
| % Poverty | 13.9 | 3.6 | 4.9 | 24.3 |
| % Undergraduate degree | 16.8 | 7.5 | 7.5 | 53.8 |
| % White | 93.6 | 6.3 | 62.7 | 98.4 |
| % Black | 2.5 | 4.5 | 0.1 | 26.7 |
| % Female | 50.3 | 1.1 | 45.7 | 53.2 |
| Metropolitan, n (%) | 46 (50%) | | | |

USD = U.S. dollar.

Differences in smoking prevalence and lung cancer incidence between counties based on county variables is shown in Table 3. Counties that had comprehensive or moderate SFOs had significantly lower rates of new lung cancer diagnosis per year (-8.36 per 100,000 population; 95% CI: -11.45 to -5.27) compared with counties with weak or no SFOs (Fig 2). Additionally, the smoking prevalence was significantly lower in counties with comprehensive or moderate SFO (-1.2%; 95% CI: -1.88% to -0.52%) compared with counties with weak or no SFOs as well (Fig 3). There was no significant difference in smoking prevalence between rural and urban counties or based on population size. However, there was a significant decrease in lung cancer incidence in counties with larger versus smaller populations (-4.5 per 100,000; 95% CI: -8.76 to -0.25) as well as increased lung cancer incidence in rural versus metropolitan locations (5.95 per 100,000; 95% CI: 1.79 to 10.12).

Smoking prevalence was higher in counties with higher poverty levels (0.3%; 95% CI: 0.13 to 0.47), higher in counties with larger percentage African-American populations (0.11%; 95% CI: 0.02 to 0.2), and lower in counties with higher percentage with college educations (-0.31%; 95% CI: -0.37 to -0.24), but there was no statistically significant difference in lung cancer incidence based on these factors. Over the study period, increasing year was also associated with

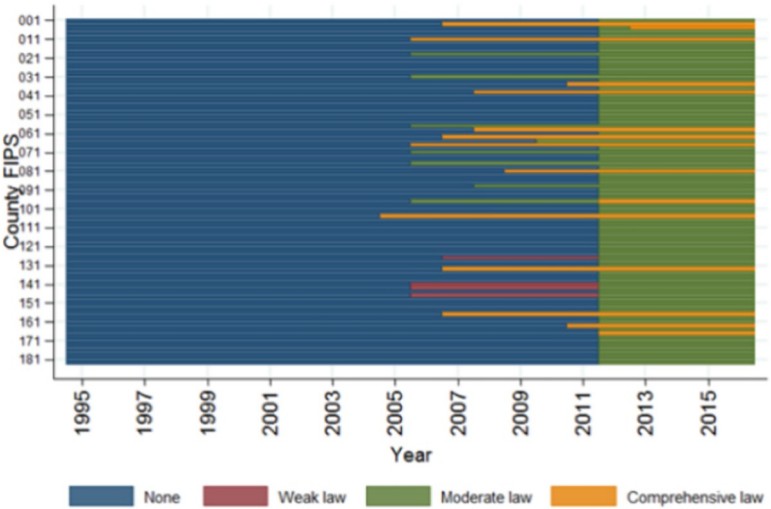

**Fig 1. Transitions of Indiana municipal smoke-free ordinances, 1995–2016.** FIPS = federal information processing standard code which identifies counties.

**Table 3. Estimated change in smoking prevalence and lung cancer incidence by county-level characteristics (1995–2016).**

| Characteristics | | Smoking prevalence (%) | | Lung cancer incidence (per 100,000) | |
|---|---|---|---|---|---|
| | | Estimates [95% CI] | p-value | Estimates [95% CI] | p-value |
| Strength of ordinance | | | | | |
| | None or Weak | Ref | | Ref | |
| | Moderate to Comprehensive | -1.2 [-1.9, -0.5] | <0.01 | -8.4 [-11.5, -5.3] | <0.01 |
| Population (log) | | -0.2 [-0.8, 0.4] | 0.45 | -4.5 [-8.8, -0.3] | 0.04 |
| Median income (USD) | | 0.08 [0.01, 0.2] | 0.09 | 0.43 [-0.4, 1.3] | 0.33 |
| % Poverty | | 0.3 [0.1, 0.5] | <0.01 | 1.5 [-0.2, 3.1] | 0.08 |
| % Undergraduate degree | | -0.3 [-0.4, -0.2] | <0.01 | -0.4 [-0.9, 0.1] | 0.08 |
| % Black | | 0.11 [0.02, 0.20] | 0.02 | 0.6 [-0.1, 1.4] | 0.1 |
| % Female | | -0.1 [-0.3, 0.1] | 0.44 | -0.2 [-1.6, 1.3] | 0.82 |
| Metropolitan | | -0.1 [-0.7, 0.6] | 0.9 | 5.9 [1.7, 10.1] | 0.01 |
| Year | | -0.2 [-0.2, -0.2] | <0.01 | 0.3 [0.1, 0.5] | <0.01 |

"Moderate to comprehensive smoke-free ordinances" change is calculated relative to none or weak smoke-free ordinance counties. "Population (log)" change is calculated based on each 10% increase in population. For continuous predictor values, circles represent estimated change based on 1% increase in corresponding characteristic (e.g., for each 1% in poverty, the smoking prevalence increases by 0.3%) except for "Median Income (USD)," which is estimated as change based on an increase of $1,000 and "year," for which change is estimated based on an increase of 1 year. For "metro", change is calculated as counties classified as metropolitan versus rural.

decreased smoking prevalence (-0.24%; 95% CI: -0.27% to -0.21%) and increasing lung cancer incidence (0.3 per 100,000; 95% CI: 0.1 to 0.51). A subgroup analysis found that lung cancer incidence was significantly higher in counties with no smoke-free ordinances (7.09 per 100,000; 95% CI: 2.75 to 11.43) and weak SFO (10.43 per 100,000; 95% CI 5.04 to 15.82) compared with comprehensive SFO counties but no statistical difference in lung cancer incidence was found between moderate and comprehensive SFO counties.

## Discussion

Despite advances in anti-smoking policies, tobacco continues to be the leading known cause of preventable morbidity and mortality in the United States. Smoke-free laws are a proven measure to decrease smoking prevalence and SHS exposure. Emerging data have also begun to demonstrate an association between stronger SFOs and improved health outcomes. A 2017 study by Hahn et al. was the first to investigate the association between SFOs and lung cancer outcomes, and found that individuals in Kentucky counties with comprehensive SFOs were 7.9% less likely to be diagnosed with lung cancer than those in counties without smoke-free protections [14]. Similarly, in our study Indiana counties with comprehensive or moderate SFO had significantly lower rates of new lung cancer diagnoses per year compared with counties with weak or no SFO (8.36 less new yearly lung cancer diagnosis per 100,000). We also found a significantly decreased smoking rate in counties with comprehensive or moderate SFO (1.2% lower smoking prevalence than counties with weak or no smoke-free ordinances). These findings suggest that stronger smoke-free ordinances are associated with fewer new lung cancer diagnoses compared with counties with weak or no smoke-free ordinances. This is one of the first studies to analyze the relationship between SFOs and lung cancer outcomes, and our study adds to the growing body of evidence linking SFOs with improved population health outcomes.

In addition to stronger SFOs, higher education levels were associated with decreased smoking prevalence among counties. Higher smoking prevalence was also found in counties with

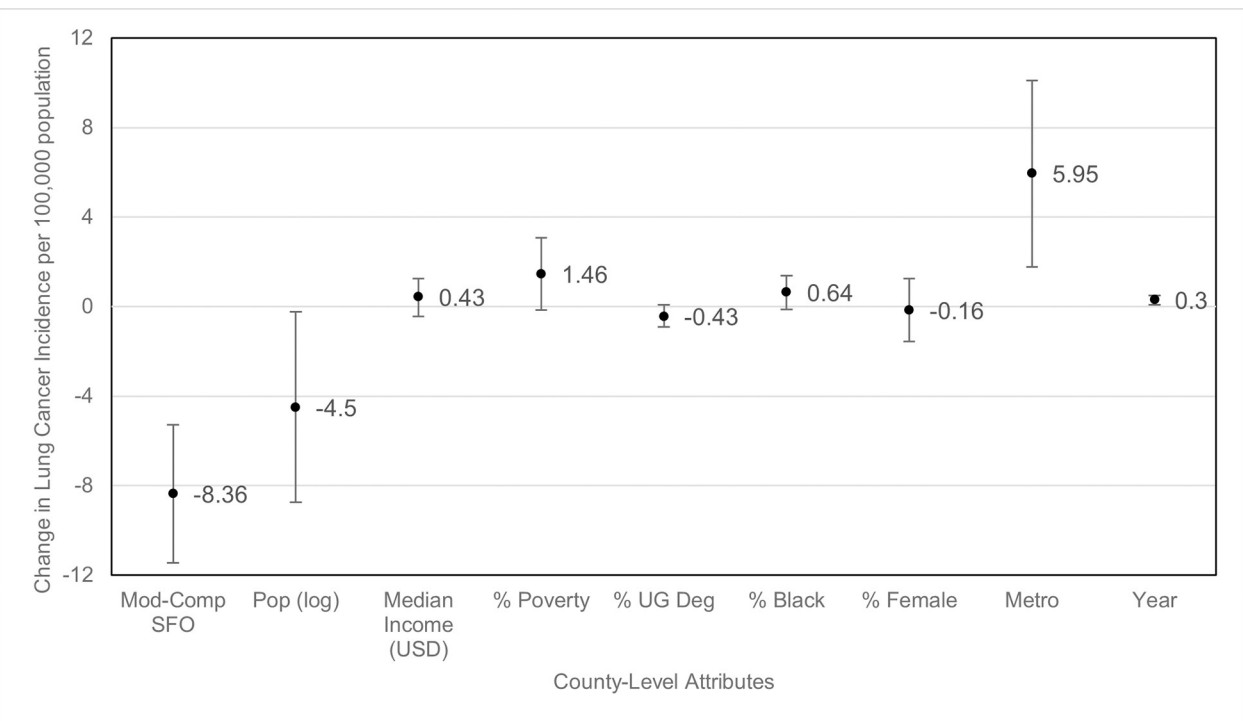

**Fig 2. Change in lung cancer incidence by county-level attributes.** Mod-Comp SFO = moderate-to-comprehensive smoke-free ordinance, pop (log) = population log, % UG Deg = percent with undergraduate degree. Circles represent estimated change in lung cancer incidence and bars represent 95% confidence intervals. "Moderate to comprehensive smoke-free ordinances" change is calculated relative to none or weak smoke-free ordinance counties. "Population (log)" change is calculated based on each 10% increase in population. For continuous predictor values, circles represent estimated change based on 1% increase in corresponding characteristic (e.g., for each 1% in poverty, the lung cancer incidence increases by 0.3%) except for "Median Income (USD)" which is estimated as change based on an increase of $1,000 and "year," for which change is estimated based on an increase of 1 year. For "metro", change is calculated as counties classified as metropolitan versus rural.

larger levels of poverty or higher African-American populations, consistent with statewide and national datasets [15]. Fewer new lung cancer diagnoses were found in counties with larger populations and located in metropolitan locations. This is most likely due to clustering of SFOs in counties with higher populations. Of the twenty most populous counties, fourteen (70%) had comprehensive or moderate SFOs by 2011. These fourteen counties also were all classified as being in metropolitan locations.

The 21-year evaluation period provided adjusted rate estimates before and after ordinance enactment for counties that passed smoke-free laws. Our data analysis method linked county-level SFO status to the corresponding lung cancer incidence for that year. The 1995 starting point is the earliest date that data were collected by the ISCR and precedes the earliest SFO by 10 years.

Interpretation of these results is subject to limitations. Our results are likely underestimating the effect of smoke-free policies on lung cancer incidence due to the extended latency period of lung cancer. Several of the smoke-free policies included in the study, including the 2012 statewide law, had been in effect for a limited time. We consider the results of this study preliminary, and additional longitudinal data will need to be reassessed for the long-term impact of more recent ordinances. Our results were also limited by lower levels of African Americans in the lung cancer demographic (7.1%) compared with the percentage of African Americans in the general population during this time period (9.1%). Further research is

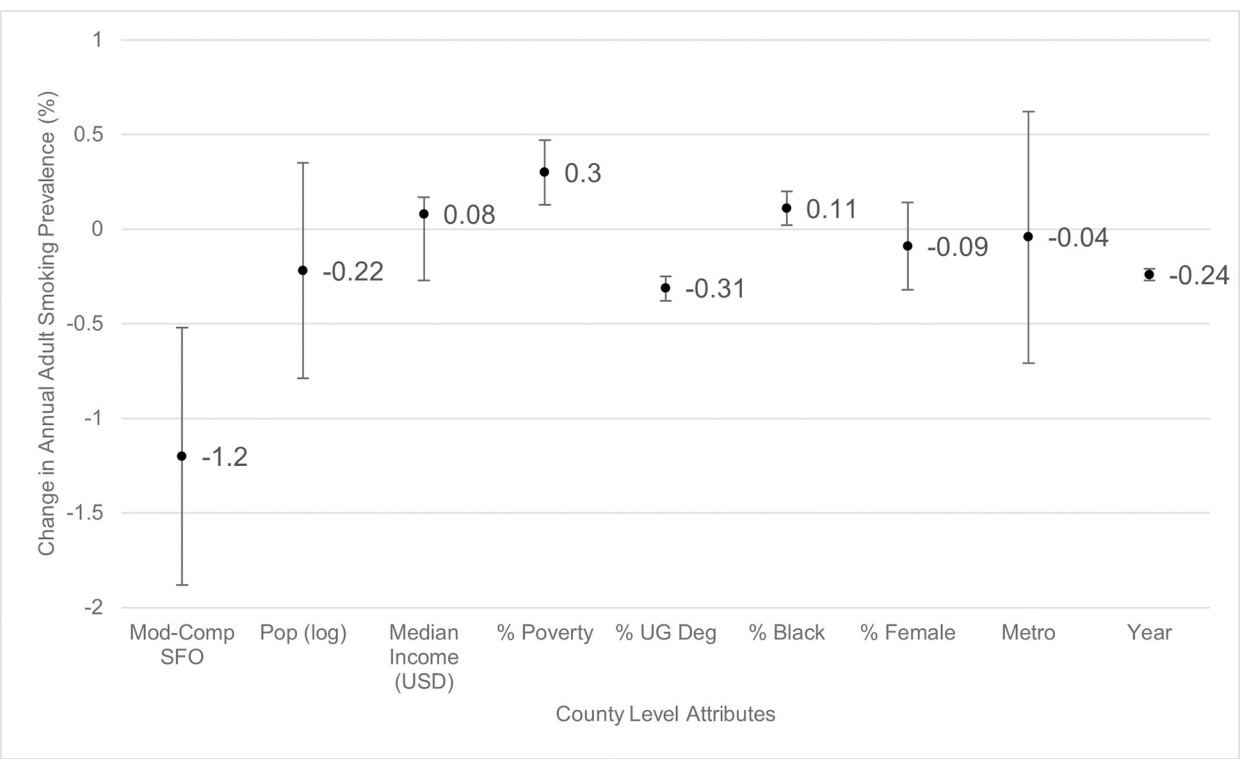

**Fig 3. Change in smoking prevalence by county-level attributes.** Mod-Comp SFO = moderate-to-comprehensive smoke-free ordinance, pop (log) = population log, % UG Deg = percent with undergraduate degree. Circles represent estimated change in smoking prevalence and bars represent 95% confidence intervals. "Moderate to comprehensive smoke-free ordinances" change is calculated relative to none or weak smoke-free ordinance counties. "Population (log)" change is calculated based on each 10% increase in population. For continuous predictor values, circles represent estimated change based on 1% increase in corresponding characteristic (e.g., for each 1% in poverty, the smoking prevalence increases by 0.3%) except for "Median Income (USD)" which is estimated as change based on an increase of $1,000 and "year," for which change is estimated based on an increase of 1 year. For "metro", change is calculated as counties classified as metropolitan versus rural.

needed to study the impact of smoke-free policies on underrepresented and vulnerable populations that exhibit higher levels of tobacco use and dependence.

A systemic review of BRFSS found moderate quality validity when comparing BRFSS tobacco use data with other national surveys [16]. The changes in BRFSS methodology in 2011 also led to an increase in smoking prevalence years after, rendering it difficult to compare trends past 2012 [17]. However, we chose to use BRFSS data from 2011–2015 because the overall trend was consistent throughout all counties in Indiana. Another limitation of the study was coding counties as having SFO if a policy only covered a city within a county. These cities were typically the most populous within the county, but it is unclear to what extent this classification accurately captured the effect of city-wide smoke-free ordinances. Additionally, survey data are subject to biases such as increased underreporting or desirability bias that could affect estimates of smoking trends.

## Conclusion

Smoke-free policies are powerful policy tools for decreasing the prevalence of smoking and improving associated public health outcomes. In Indiana, counties with comprehensive or moderate smoke-free ordinances were found to have fewer new lung cancer diagnosis per year compared to counties with weak or no smoke-free ordinances. Strengthening smoke-free ordinances is paramount to decreasing lung cancer incidence.

## Supporting information

**S1 File.**
(XLSX)

## Acknowledgments

We thank the Indiana State Cancer Registry and the Indiana Tobacco Prevention and Cessation Committee for providing the data for this study.

## Author Contributions

**Conceptualization:** Ryan H. Nguyen, Laura B. Vater, DuyKhanh P. Ceppa.

**Data curation:** Ryan H. Nguyen.

**Methodology:** Ryan H. Nguyen, Laura B. Vater, Lava R. Timsina, DuyKhanh P. Ceppa.

**Resources:** Katelin Rupp, Keylee Wright, Miranda H. Spitznagle, Brandy Paul.

**Supervision:** Nasser H. Hanna, Patrick J. Loehrer, DuyKhanh P. Ceppa.

**Validation:** Lava R. Timsina, Katelin Rupp, Keylee Wright, Miranda H. Spitznagle, Brandy Paul.

**Writing – original draft:** Ryan H. Nguyen, Laura B. Vater, Lava R. Timsina, DuyKhanh P. Ceppa.

**Writing – review & editing:** Ryan H. Nguyen, Laura B. Vater, Lava R. Timsina, Gregory A. Durm, Shadia I. Jalal, Lisa Carter-Harris, Karen S. Hudmon, Nasser H. Hanna, Patrick J. Loehrer, DuyKhanh P. Ceppa.

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
