## [Decision Letter · Decision Letter 0]

3 Mar 2021

PONE-D-21-01837

Impact of smoke-free ordinance strength on smoking prevalence and lung cancer incidence

PLOS ONE

Dear Dr. Nguyen,

Thank you for submitting your manuscript to PLOS ONE. After careful consideration, we feel that it has merit but does not fully meet PLOS ONE’s publication criteria as it currently stands. Therefore, we invite you to submit a revised version of the manuscript that addresses the points raised during the review process.

We look forward to receiving your revised manuscript.

Kind regards,

Stanton A. Glantz

Academic Editor

PLOS ONE

Journal Requirements:

2. We noticed your Abstract has previously been published at the following publication and appears to be currently under copyright there:

- https://ascopubs.org/doi/10.1200/JCO.2019.37.15_suppl.6578

Before we can proceed, please clarify whether the authors have received written permission from  American Society of Clinical Oncology (ASCO) to publish this content specifically under the CC BY 4.0 license and upload the granted permission to the manuscript as a supporting information file. Please note that RightsLink permission forms often impose use restrictions that are incompatible with our CC BY 4.0 license, and we are therefore unable to accept these permissions.

For this reason, we strongly recommend contacting copyright holders with the PLOS ONE Request for Permission form. (http://journals.plos.org/plosone/s/file?id=7c09/content-permission-form.pdf)

3. In the ethics statement in the manuscript and in the online submission form, please provide additional information about the patient records/samples used in your retrospective study.

Specifically, please ensure that you have discussed whether all data/samples were fully anonymized before you accessed them.

5. Thank you for stating the following financial disclosure: 'No'

6.  Please include a separate caption for each figure in your manuscript.

7. Please ensure that you refer to Figures 2 and 3 in your text as, if accepted, production will need this reference to link the reader to the figure.

8. Please include your tables as part of your main manuscript and remove the individual files. Please note that supplementary tables should be uploaded as separate "supporting information" files.

Reviewers' comments:

Reviewer's Responses to Questions

**Comments to the Author**

1. Is the manuscript technically sound, and do the data support the conclusions?

Reviewer #1: Yes

2. Has the statistical analysis been performed appropriately and rigorously? 

Reviewer #1: Yes

3. Have the authors made all data underlying the findings in their manuscript fully available?

Reviewer #1: No

4. Is the manuscript presented in an intelligible fashion and written in standard English?

Reviewer #1: Yes

5. Review Comments to the Author

Reviewer #1: This paper is well undertaken and well written replication of results in Indiana reported earlier by Hahn and colleagues for Kentucky. This earlier study is appropriately referenced

In the first sentence of the last paragraph of the introduction, the authors claim that SFOs are one of the most effective public policy means of decreasing smoking prevalence and SHS exposure. While the reference they use is appropriate, it is not sufficient for such a strong statement. Suggest the authors add a reference to an official report such as:

IARC Handbooks of Cancer Prevention, Tobacco Control, Vol. 13: Evaluating the effectiveness of smoke-free policies (2009: Lyon, France)

Another relevant reference for this would be Azagba S, Shan L, Latham K. County Smoke-Free Laws and Cigarette Smoking Among U.S. Adults, 1995-2015. Am J Prev Med. 2020 Jan;58(1):97-106.

6. PLOS authors have the option to publish the peer review history of their article (what does this mean?). If published, this will include your full peer review and any attached files.

Reviewer #1: **Yes: **John P Pierce

---

## [Author Response · Author response to Decision Letter 0]

15 Mar 2021

Thank you for submitting your manuscript to PLOS ONE. After careful consideration, we feel that it has merit but does not fully meet PLOS ONE’s publication criteria as it currently stands. Therefore, we invite you to submit a revised version of the manuscript that addresses the points raised during the review process.

We look forward to receiving your revised manuscript.

Kind regards,

Stanton A. Glantz

Academic Editor

PLOS ONE

Journal Requirements:

https://hes32-ctp.trendmicro.com:443/wis/clicktime/v1/query?url=https%3a%2f%2fjournals.plos.org%2fplosone%2fs%2ffile%3fid%3dwjVg%2fPLOSOne%5fformatting%5fsample%5fmain%5fbody.pdf&umid=31a5cdc5-6086-4765-8289-19fb71870268&auth=85c5a955287d1e42fab58bed777dfa626e5ad059-9e50d51f543c7be356a8c1b6a748fb3053e3adb6 and

We have updated the manuscript to meet PLOS ONE style requirements.

2. We noticed your Abstract has previously been published at the following publication and appears to be currently under copyright there:

- https://hes32-ctp.trendmicro.com:443/wis/clicktime/v1/query?url=https%3a%2f%2fascopubs.org%2fdoi%2f10.1200%2fJCO.2019.37.15%5fsuppl.6578&umid=31a5cdc5-6086-4765-8289-19fb71870268&auth=85c5a955287d1e42fab58bed777dfa626e5ad059-a18a4906472abc171a3c013e63cfae491a866e9d

Before we can proceed, please clarify whether the authors have received written permission from American Society of Clinical Oncology (ASCO) to publish this content specifically under the CC BY 4.0 license and upload the granted permission to the manuscript as a supporting information file. Please note that RightsLink permission forms often impose use restrictions that are incompatible with our CC BY 4.0 license, and we are therefore unable to accept these permissions.

For this reason, we strongly recommend contacting copyright holders with the PLOS ONE Request for Permission form. (https://hes32-ctp.trendmicro.com:443/wis/clicktime/v1/query?url=http%3a%2f%2fjournals.plos.org%2fplosone%2fs%2ffile%3fid%3d7c09%2fcontent%2dpermission%2dform.pdf&umid=31a5cdc5-6086-4765-8289-19fb71870268&auth=85c5a955287d1e42fab58bed777dfa626e5ad059-b15d5c7d04b59a9faaea6d285b99edeb8bc59587)

We were unable to obtain the specific CC BY 4.0 license for this abstract, as such we have modified the abstract to no longer require obtaining copyright from ASCO for the abstract being submitted in this manuscript to PLOS ONE.

3. In the ethics statement in the manuscript and in the online submission form, please provide additional information about the patient records/samples used in your retrospective study.

Specifically, please ensure that you have discussed whether all data/samples were fully anonymized before you accessed them.

We have updated the ethics statement to indicate all data/samples were fully anonymized before we accessed them. 

4. We note that you have indicated that data from this study are available upon request. PLOS only allows data to be available upon request if there are legal or ethical restrictions on sharing data publicly. For information on unacceptable data access restrictions, please see https://hes32-ctp.trendmicro.com:443/wis/clicktime/v1/query?url=http%3a%2f%2fjournals.plos.org%2fplosone%2fs%2fdata%2davailability%23loc%2dunacceptable%2ddata%2daccess%2drestrictions&umid=31a5cdc5-6086-4765-8289-19fb71870268&auth=85c5a955287d1e42fab58bed777dfa626e5ad059-6bd67d2e302e59d8d1d9f13691bd4da8c206a046.

b) If there are no restrictions, please upload the minimal anonymized data set necessary to replicate your study findings as either Supporting Information files or to a stable, public repository and provide us with the relevant URLs, DOIs, or accession numbers. Please see https://hes32-ctp.trendmicro.com:443/wis/clicktime/v1/query?url=http%3a%2f%2fwww.bmj.com%2fcontent%2f340%2fbmj.c181.long&umid=31a5cdc5-6086-4765-8289-19fb71870268&auth=85c5a955287d1e42fab58bed777dfa626e5ad059-f8d40643eb43b63efe44ba4fb1211cd57abaadeb for guidelines on how to de-identify and prepare clinical data for publication. For a list of acceptable repositories, please see https://hes32-ctp.trendmicro.com:443/wis/clicktime/v1/query?url=http%3a%2f%2fjournals.plos.org%2fplosone%2fs%2fdata%2davailability%23loc%2drecommended%2drepositories&umid=31a5cdc5-6086-4765-8289-19fb71870268&auth=85c5a955287d1e42fab58bed777dfa626e5ad059-4f8ea64c6c40d88f0cb5822e7e32889eb4ec41ef.

We have provided a minimal anonymized data set. 

 5. Thank you for stating the following financial disclosure: 'No'

Please clarify the sources of funding (financial or material support) for your study. List the grants or organizations that supported your study, including funding received from your institution.

State what role the funders took in the study. If the funders had no role in your study, please state: “The funders had no role in study design, data collection and analysis, decision to publish, or preparation of the manuscript.”

If any authors received a salary from any of your funders, please state which authors and which funders.

If you did not receive any funding for this study, please state: “The authors received no specific funding for this work.”

We have updated our statement to “The authors received no specific funding for this work.”

6. Please include a separate caption for each figure in your manuscript.

We have included a separate caption for each figure in our manuscript. 

7. Please ensure that you refer to Figures 2 and 3 in your text as, if accepted, production will need this reference to link the reader to the figure.

We have referred to figures 2 and 3 in the text. 

8. Please include your tables as part of your main manuscript and remove the individual files. Please note that supplementary tables should be uploaded as separate "supporting information" files.

We have included the tables as part of the manuscript. Per the email from Jazmin Toth (PONE-D-21-01837R1) on 3/15/21, I have re-included copies of the tables as separate files. 

Reviewers' comments:

Reviewer's Responses to Questions

Comments to the Author

1. Is the manuscript technically sound, and do the data support the conclusions?

Reviewer #1: Yes

2. Has the statistical analysis been performed appropriately and rigorously?

Reviewer #1: Yes

3. Have the authors made all data underlying the findings in their manuscript fully available?

Reviewer #1: No

4. Is the manuscript presented in an intelligible fashion and written in standard English?

Reviewer #1: Yes

5. Review Comments to the Author

Reviewer #1: This paper is well undertaken and well written replication of results in Indiana reported earlier by Hahn and colleagues for Kentucky. This earlier study is appropriately referenced

In the first sentence of the last paragraph of the introduction, the authors claim that SFOs are one of the most effective public policy means of decreasing smoking prevalence and SHS exposure. While the reference they use is appropriate, it is not sufficient for such a strong statement. Suggest the authors add a reference to an official report such as:

IARC Handbooks of Cancer Prevention, Tobacco Control, Vol. 13: Evaluating the effectiveness of smoke-free policies (2009: Lyon, France)

Another relevant reference for this would be Azagba S, Shan L, Latham K. County Smoke-Free Laws and Cigarette Smoking Among U.S. Adults, 1995-2015. Am J Prev Med. 2020 Jan;58(1):97-106.

We have updated this citation with the two references provided, thank you for your suggestion. 

6. PLOS authors have the option to publish the peer review history of their article (what does this mean?). If published, this will include your full peer review and any attached files.

Do you want your identity to be public for this peer review? For information about this choice, including consent withdrawal, please see our Privacy Policy.

Reviewer #1: Yes: John P Pierce

---

## [Editor Report · Decision Letter 1]

5 Apr 2021

Impact of smoke-free ordinance strength on smoking prevalence and lung cancer incidence

PONE-D-21-01837R1

Dear Dr. Nguyen,

We’re pleased to inform you that your manuscript has been judged scientifically suitable for publication and will be formally accepted for publication once it meets all outstanding technical requirements.

Kind regards,

Stanton A. Glantz

Academic Editor

PLOS ONE
---

## [Editor Report · Acceptance letter]

8 Apr 2021

PONE-D-21-01837R1 

Impact of smoke-free ordinance strength on smoking prevalence and lung cancer incidence 

Dear Dr. Nguyen:

I'm pleased to inform you that your manuscript has been deemed suitable for publication in PLOS ONE. Congratulations! Your manuscript is now with our production department. 

Kind regards, 

on behalf of

Professor Stanton A. Glantz 

Academic Editor

PLOS ONE